# Stem Cell Origin of Cancer: Clinical Implications for Cancer Immunity and Immunotherapy

**DOI:** 10.3390/cancers15225385

**Published:** 2023-11-13

**Authors:** Shi-Ming Tu, Ahmet Murat Aydin, Sanjay Maraboyina, Zhongning Chen, Sunny Singh, Neriman Gokden, Timothy Langford

**Affiliations:** 1Division of Hematology and Oncology, University of Arkansas for Medical Sciences, Little Rock, AR 72205, USA; zchen3@uams.edu (Z.C.); srsingh@uams.edu (S.S.); 2Department of Urology, University of Arkansas for Medical Sciences, Little Rock, AR 72205, USA; maydin@uams.edu (A.M.A.); tlangford@uams.edu (T.L.); 3Department of Radiation Oncology, University of Arkansas for Medical Sciences, Little Rock, AR 72205, USA; smaraboyina@uams.edu; 4Department of Pathology, University of Arkansas for Medical Sciences, Little Rock, AR 72205, USA; gokdenneriman@uams.edu

**Keywords:** cancer vaccine, immunotherapy, cancer stem cell, hyper-progression, heterogeneity, microenvironment, autoimmunity, microbiome

## Abstract

**Simple Summary:**

According to the stem cell theory of cancer, we should be judicious with immunotherapy in cancer care. When we do not realize that cancer originates from a stem cell and has stem-ness capabilities, immunotherapy could be gratifying for the wrong reasons and challenging with a cautionary tale.

**Abstract:**

A simple way to understand the immune system is to separate the self from non-self. If it is self, the immune system tolerates and spares. If it is non-self, the immune system attacks and destroys. Consequently, if cancer has a stem cell origin and is a stem cell disease, we have a serious problem and a major dilemma with immunotherapy. Because many refractory cancers are more self than non-self, immunotherapy may become an uphill battle and pyrrhic victory in cancer care. In this article, we elucidate cancer immunity. We demonstrate for whom, with what, as well as when and how to apply immunotherapy in cancer care. We illustrate that a stem cell theory of cancer affects our perspectives and narratives of cancer. Without a pertinent theory about cancer’s origin and nature, we may unwittingly perform misdirected cancer research and prescribe misguided cancer treatments. In the ongoing saga of immunotherapy, we are at a critical juncture. Because of the allure and promises of immunotherapy, we will be treating more patients not immediately threatened by their cancer. They may have more to lose than to gain, if we have a misconception and if we are on a wrong mission with immunotherapy. According to the stem cell theory of cancer, we should be careful with immunotherapy. When we do not know or realize that cancer originates from a stem cell and has stem-ness capabilities, we may cause more harm than good in some patients and fail to separate the truth from the myth about immunotherapy in cancer care.

## 1. Introduction


*It is difficult to separate, at times, the myth from the truth.*
—Bob Kane

When we know the origin of something, we will know many important things about that something. Knowing the origin of cancer will help us separate the myth from the truth about cancer. We will know how to conduct enhanced cancer research and provide improved cancer care.

Nowadays, there is controversy about a stem cell theory of cancer [1,2,3,4]. A stem cell theory may provide us with a comprehensive understanding and knowledge about the origin and nature of cancer [5,6]. It may be the elusive unified theory of cancer that elucidates the origin of all cancer hallmarks, including heterogeneity and immune evasion [5,6]. It embraces the genomics and epigenomics of cancer. It unites various compartments, different components, and the microenvironment of cancer. Importantly, a unified theory of cancer may empower us to treat the whole rather than just a part of cancer. It advocates multimodal therapy over targeted therapy to upgrade cancer care. It advances integrated medicine over precision medicine to fulfill cancer cure.

Currently, immunotherapy is immensely popular and overwhelmingly prevalent in cancer care. Although its scientific merits are undoubted and clinical values unquestioned, there are some inconvenient truths about its promises and limitations that need to be addressed, so that we do not befall to herd mentality or peer pressure in our scientific endeavors and clinical odysseys. We hope that some burning questions about and an alternative view of immunotherapy will invite informed debate and draw objective attention to its putative shortcomings and drawbacks that will further advance and enhance its clinical utility and benefits in cancer care.

In this perspective, we use the example of cancer immunity and immunotherapy to illustrate the clinical implications of a unified theory of cancer and the perils of scientific research conducted without guidance of the scientific method. We demonstrate that a stem cell theory of cancer affects our perspectives and narratives of cancer. It determines how we articulate our questions and formulate our hypotheses to answer those questions. It dictates how we design our experiments to test the hypotheses and interpret the results derived from those experiments. We need to be cognizant that a pertinent cancer theory regarding the origin of cancer has enormous impact and implications on the directions of cancer research and on the destinations of cancer care.

## 2. A Systemic Problem

To conquer cancer is to control it, if not to cure it. This requires that we know what cancer really is—down to its very core and crux. When we scratch the surface but do not search deeper, we may never find its seeds or see its roots. Furthermore, when we focus on the parts but miss the whole, we may briefly and barely control cancer. Importantly, when we have a misconception and misunderstanding about cancer’s origin and nature, we may conquer an occasional cancer for the wrong reasons.

When cancer is a genetic aberration, we correct the aberration. When it is a metabolic malfunction, we resolve the malfunction. When it is a microenvironment imbalance, we steady the imbalance. When it is an immune defect, we repair the defect.

However, if cancer is all of the above, i.e., a systemic problem, then treating just the genetic aberration, metabolic malfunction, microenvironment imbalance, or immune defect, in an interconnected and interrelated network of cancer is likely to be insufficient and unsatisfactory. When we are lucky enough that a particular “targeted therapy” or “precision medication” is effective and safe, any clinical benefit is likely to be modest rather than monumental and incremental rather than exponential.

It is evident that there is a close relationship between oncogene and pro-oncogene and a close resemblance between cancer stem cells and normal stem cells. However, it is not yet apparent if this relationship is pivotal to solving the puzzle of cancer and if a resemblance is the missing key to unlocking the origin of cancer. When a proto-oncogene is already existent, perhaps an oncogene is a reinvention. When a normal stem cell is always present, perhaps a cancer stem cell is a recapitulation.

When it concerns drug development and drug approval by the Federal Drug Administration (FDA), perhaps statistical considerations of clinical outcomes are good enough for our purposes. However, when we consider therapy development [3] and optimal patient care, we need more than just availability of effective and safe drugs: we need a better understanding about the origin and nature of cancer and a better strategy as to why, whom, which, what, when, where, and how to apply those treatments (Table 1) in an effort to maximize patient benefit for the right reasons.

## 3. The Immune System

When we mull over immunity, multicellularity is a reality. When we deal with multicellularity, immunity is a necessity. We need to discern self from non-self. We need to distinguish friends from foes.

Another prerequisite for multicellularity is heterogeneity. There is hierarchy and specialty. All cells have a role and function. They are different, not equal. Some cells are permanent and indispensable. Others are transient and disposable.

Stem cells are destined to be permanent and indispensable for a good reason. They are the lifeline. They are the seed of life. To ensure perpetuity, they renew and regenerate, migrate and hibernate.

In contrast, differentiated cells are designed to serve certain purposes for the good of a whole community of cells, like the leaves and flowers do. They sprout and blossom. They appear and disappear.

Although both stem cells and differentiated cells are vital in a multicellular organism, there is a difference in the priority and in the methods of preserving the whole and the parts.

Consequently, when there is damage, it is imperative that we repair the damage in a stem cell compared with that in a differentiated cell. If the damage were irreparable, it would be easier and safer to remove and replace a differentiated cell than a stem cell.

Therefore, it makes sense that the immune system is more likely to remove and replace a defective differentiated cell than an aberrant stem cell. A stem cell is immune-privileged for a good reason. It is equipped to elude the radar and evade the scrutiny of immune surveillance. After all, we cannot afford to eliminate a fountain and the spring of all cells and eradicate our very self.

Understandably, normal stem cells are protected from the rigors of a normal immune system by virtue of its abundant immune checkpoints, its lack of MHC class I molecules, its alliance with HLA-G antigen and regulatory T cells and truce with natural killer cells and cytotoxic T cells [56]. Unsurprisingly, there will be a predominance of inhibitory immune cells and a preponderance of inhibitory immune factors in a normal stem-ness microenvironment [57].

## 4. Cancer Immunity

Unfortunately, if cancer has a stem cell origin and is a stem cell disease, then whether the cancer stem cell is self or non-self not only presents us with some perturbing and pressing questions, but also confronts us with some perplexing and profound challenges when it concerns cancer immunity and cancer care. If the immune system spares normal stem cells, then it will also exempt cancer stem cells* (Appendix: *a small subpopulation of cells within tumors capable of self-renewal, differentiation, and tumorigenicity when transplanted into an animal host. After all, cancer stem cells derive from normal stem cells. They mirror, if not mimic one another (Figure 1)).

No matter how we envision or imagine it, when a cancer stem cell hijacks, sabotages, reprograms, or inherits the innate plurality and diversity, resiliency and versatility, immune privileges and exemptions of a normal stem cell, we will have a problem controlling or curing cancer with immunotherapy, as we do with other conventional or novel therapeutic modalities.

Unless and until we have a unified theory of cancer, immunotherapy alone like surgery, radiotherapy, and chemotherapy may only conquer certain parts or control certain aspects of a cancer. It is another useful therapeutic option, but unlikely to be the proverbial panacea. The same questions and challenges apply: why, whom, which, what, when, where, and how do we apply it in order to optimize cancer care (Table 1). Without a proper theory about cancer’s origin and nature, we are vulnerable to performing misdirected cancer research and prescribing misguided cancer treatments.

### 4.1. Cancer Vaccines (Why?)

In many respects, an epitome of immunotherapy is vaccine therapy. However, vaccines have been generally ineffective in cancer therapy. An inconvenient truth is why? Perhaps we tend to forget or ignore a plethora of negative cancer vaccine trials. We are inclined to dismiss if not deny negative data. But negative studies may have important lessons, messages, and warnings for us all, too. The reason cancer vaccines have not (yet) delivered on many fronts may be more than imperfect drugs, improper study designs, or inadequate technical know-hows—the problems with cancer vaccines may be more fundamental than accidental and radical than nominal—than we think or realize.

#### 4.1.1. Prostate Cancer

Consider the case of sipuleucel-T, the first cancer vaccine ever approved by the FDA in 2010 for patients with asymptomatic or minimally symptomatic metastatic castration-resistant prostate cancer (CRPCa). Statistically, patients who received the vaccine experienced a significant overall survival benefit of about 4 months compared with those who did not [7,8].

Clinically, it remains unclear whether sipuleucel-T actually provides any meaningful clinical benefit. There were no durable remissions or long-term disease-free survivors. Almost all patients had rising PSA levels and growing tumors during and right after treatment. About 10% of patients may even experience rapid progression of disease and early death [58]. There is debate whether the results of sipuleucel-T represent a statistical sleight of hand, or a clinical fluke of nature [59]. Perhaps we will never know.

For those asymptomatic or minimally symptomatic CRPCa patients with limited bone or soft tissue metastasis who received sipuleucel-T, is the reported overall survival of 25 months better than expected? If the treatment somehow adversely affected or potentially harmed the control group, then the purported results could be statistically significant but clinically ambiguous. In addition, why was the overall survival of the control group not improved after crossover to sipuleucel-T?

Intriguingly, two phase 3 trials showed that a supposedly superior cell-based vaccine, GVAX, did not provide any overall survival improvement [9,10]. In fact, interim analysis of VITAL-2 suggested an increased risk of death with GVAX (data not published). Similarly, despite promising even impressive preliminary results, a virus-based vaccine, PROSTVAC, failed to detect any survival advantage in a global phase 3 trial enrolling 1297 patients with metastatic CRPCa [11].

#### 4.1.2. Melanoma

In principle, a vaccine should be given preemptively before the disease occurs or proactively to prevent the disease altogether. In practice, the vaccine should be tested in a tumor, such as melanoma, known to be highly immunogenic and has the best track record of responding to immunotherapy. Ideally, the vaccine should be given when the disease burden is low and the tumor relatively naïve. If these premises are true, then the optimal time and way to administer a cancer vaccine is in an adjuvant setting for the treatment of high-risk but low-stage melanoma.

Unfortunately, all adjuvant trials testing peptide vaccines, ganglioside vaccines, and whole cells/cell lysates for the treatment of melanoma have so far been utter disappointments or frank failures. For example, results from a phase 3 trial using a peptide vaccine did not show benefit when given in an adjuvant setting for high-risk resected melanoma patients [12]. The multi-epitope peptide vaccine was only immunologically recognized in the context of human leukocyte antigen (HLA)-A2. But there was no significant difference in overall survival between HLA-A2-positive and -negative patients.

Similarly, an Australian study using vaccinia viral lysates in high-risk subjects following resection failed to show a statistically significant increase in relapse-free survival time [13]. The Melacine vaccine trial initially showed some promise, but failed to sustain it [14]. A phase 3 trial for resected stage III/IV melanoma using the polyvalent vaccine Canvaxin versus BCG vaccination showed that patients who received Canvaxin had worse disease-free survival and overall survival times [15]. The E1694 trial that tested adjuvant GM2-KLH21 vaccination was shown to be ineffective and could even be detrimental in stage II melanoma patients [16,17]. A randomized phase 2 trial (DERMA) tested adjuvant therapy where MAGE-A3 protein did not reach its primary endpoint of relapse-free survival [18].

#### 4.1.3. Kidney Cancer

Perhaps the next best malignancy after melanoma for the development of cancer vaccines is renal cell carcinoma (RCC). After all, not too long ago, the best immunotherapy, namely, high-dose interleukin-2, did provide durable complete responses in <10% of patients with metastatic melanoma or kidney cancer [60,61].

However, a cancer vaccine (Trovax) containing the tumor-associated antigen (5T4) delivered by the pox virus vector, a modified vaccinia virus Ankara (MVA), did not improve overall survival in patients with metastatic RCC when combined with either sunitinib, IL-2, or IFN-alpha [19].

The multipeptide cancer vaccine IMA901 also did not improve survival when added to sunitinib in patients with RCC [20]. IMA901 consists of nine different HLA class I-binding tumor associated peptides. In the phase 3 open-labeled IMPRINT trial, the IMA901 group received an average of 9.3 vaccinations, with 80% of patients receiving all 10 scheduled vaccinations with IMA901 and GMCSF. After a median of 33.27 months, 50% of patients who received sunitinib + IMA901 and 40% of those who received sunitinib monotherapy had died. Median PFS from randomization was 15.22 vs. 15.12 months, respectively. There was no association between T-cell response and clinical outcome.

Perhaps the chance of benefit from the cancer vaccine may be greater for RCC patients with a low-volume tumor but high risk for recurrence in an adjuvant setting rather than with metastatic disease (similar to the rationale and logic depicted for melanoma). However, autologous irradiated tumor cells combined with BCG did not improve disease-free or overall survival [21]. Similarly, Vitespen (also known as Oncophage or HSPPC-96), a heat-shock protein (glycoprotein 96)–peptide complex that is purified ex vivo from an individual patient’s tumor cells failed to show broad activity in randomized clinical trials despite encouraging results in select patients [22,23].

### 4.2. Patient Selection (Whom?)

Identifying the right patients and providing them with the right treatment is a holy grail in patient care. When immunotherapy is more effective for patients whose cancer shows microsatellite instability (MSI) [24] or harbors high tumor mutation burden (TMB > 10 mutations/megabase) [25] in a variety of cancers, we may have reached the pinnacle of precision medicine.

Or perhaps this crucial clinical observation reveals a universal truth about cancer: it alludes to a unified theory of cancer in which a common theme removes divisions and a common mechanism crosses borders among a multitude of disparate cancers. And we should aim even further and higher for a paragon of integrated medicine in cancer care.

For example, we already know that normal human stem cells possess highly efficient DNA repair mechanisms that become less efficient upon differentiation [62,63].

Casorelli et al. demonstrated higher expression as well as functional activity and efficiency of mismatch repair (MSH2, MSH6, MLH1, PMS2), base excision repair (AAG, APEX), and O(6)-methylguanine DNA methyltransferase in CD34+ stem cells compared with the terminally differentiated CD34-cells [62].

The rate of removal of DNA adducts, the resealing of repair gaps and the resistance to DNA-reactive drugs were higher in stem (CD34+ 38-) than in mature (CD34-) or progenitor (CD34+ 38+) cells from the same individual [63].

Furthermore, stem cells utilize anaerobic glycolytic metabolism (rather than mitochondrial or oxidative metabolism), which reduces oxidative stress and DNA damage as well as cellular injury in general [64,65].

Stem cells have another mechanism to minimize the chance of replication error: being quiescent, remaining at the G0 phase of the cell cycle, and having a short cell cycle (G1, S, G2, and M) [66].

The unique G1 kinetics and partial deficiency in G1/S checkpoint allow damaged stem cells to progress into S, which amplifies the DNA damage, leading to cell death. Perhaps this is one reason embryonal carcinoma is exquisitely sensitive to the cytotoxic effects of chemotherapy.

Importantly, when DNA damage repair is impaired, stem cells undergo senescence, cell death, or differentiation, in order to avoid the propagation of potentially harmful genetic mutations and genomic alterations to their offspring cells.

Consequently, in cases of severe or excessive DNA damage, p53 induces apoptosis or senescence. Furthermore, activation of p53 suppresses pluripotency genes, such as Nanog, allowing differentiation to proceed [67,68].

Otherwise, when control of DNA damage is in disarray, stem cells are prone to causing errors in asymmetric division and ending up with genetic instability.

Therefore, we speculate that progenitor stem-like cells are more likely to succumb to defective asymmetric division and chromosomal aneuploidy but less likely to develop into hypermutated MSI tumors compared with progeny differentiated cancer cells.

Interestingly, there are two subtypes of colorectal cancer. Patients with hypermutated, diploid, MSI colorectal cancer carry a better prognosis compared to those with non-hypermutated, aneuploidy, and microsatellite stable (MSS) colorectal cancer. The former tumor subtype is associated with tumor infiltrating lymphocytes (TIL) [69], and better response to immunotherapy. We postulate that the better prognostic and predictive immune-phenotype may in fact be related to its unique stem-ness origin and nature, which encompass MSI vs. MSS status and TIL presence vs. absence.

### 4.3. CPIs Are Not Equal (Which?)

A mantra of immunotherapy is that it manipulates the native and a naive immune system to improve identification and enhance the elimination of cancer. However, a salient question is which immunotherapy should we use to target a particular cancer subtype and which cancer subtype should we design an immunotherapy to target and treat it. Unfortunately, without a proper scientific theory about the origin and nature of cancer, we may not realize that the basic concept of immunotherapy may be misconceived and the development of some immunotherapy misguided.

Although checkpoint inhibitors (CPI) targeting PD1/PDL1 have been generally effective for a variety of malignancies, the same is not true for several others, such as those targeting IDO1, TIGIT, or CD47. The temptation is to add or combine various CPI. However, if the whole idea about immunotherapy in cancer care is fundamentally flawed, then we have a problem with the idea, if not with the treatment. If anti-PD1/PDL1 is effective because it is tumor-targeting rather than immune-modulating, then immunotherapy in cancer care is effective for the wrong reasons. Importantly, if many of the other CPI, such as those targeting IDO1, TIGIT, or CD47, are either minimally effective or not effective at all, then we may be wasting our precious resources, energy, and time.

Perhaps we forget that even under the best circumstances, any clinical benefit from immunotherapy is likely to be limited or modest. This is exactly what had happened with high-dose interleukin-2 (IL-2) not too long ago, when <10% of patients with metastatic melanoma experienced a durable complete response [60]. Unfortunately, recent efforts to revive an engineered IL-2 pathway agonist (NKTR-214 or bempegaldesleukin) in patients with untreated RCC or urothelial carcinoma have also been rather unsatisfactory or outright disappointing [26].

Perhaps true immunotherapy that is immune-modulating rather than tumor-targeting is less effective for a telling reason. So far, ipilimumab (anti-CTLA-4) by itself provides marginal clinical benefit [27]. In addition, tremelimumab alone or in combination (vs. anti-PDL1 alone) has been either marginally effective or relatively ineffective [28,29,30]. Other highly touted treatments, e.g., anti-IDO1, anti-TIGIT, and anti-CD47, that modulate the immune system but do not target a pertinent cancer compartment or components, such as cancer stem cells (CSC), have also been generally under-performing [31,32].

In contrast, anti-PD-L1/PD-1 is clearly efficacious and beneficial for the treatment of a variety of cancers [33,34,35,36,37]. Among the myriad immune checkpoints, PD-L1 is one of the most intriguing, because it is also expressed on normal stem cells, such as mesenchymal stem cells [70,71]. If cancer has a stem cell origin, then unlike other immune CPI, anti-PD1/L1 has a bona fide anti-cancer (e.g., anti-CSC) capability beside immune-modulating activity.

For example, PD-L1 modulates the epithelial–mesenchymal transition (EMT) and the CSC-like phenotype [72]. Induction of EMT may upregulate expression of PD-L1 and other immune checkpoints in claudin-low breast cancer [73], NSCLC [74], and RCC [75].

In addition, OCT4 signaling and upregulation of the EMT-inducer ZEB1 induces PD-L1 expression. PD-L1 is co-amplified along with other stem-ness factors, such as MYC, SOX2, n-cadherin, and SNAI1 [76].

Furthermore, EMT transcriptionally induces Stat-3 (STT3) through beta-catenin, and subsequent STT3-dependent PD-L1 stabilization and upregulation in CSC more than non-CSC [77].

Therefore, PD-L1 is linked to EMT through stem-ness. Anti-PD1/L1 is a genuine anti-cancer (e.g., anti-CSC) treatment beside, or perhaps beyond, immune-modulating therapy.

In other words, immune checkpoints are not equal: the ones (e.g., PD-L1) that protect CSC more than non-CSC may have more clinical relevance (e.g., adjuvant vs. neoadjuvant), while those that protect non-CSC, non-cancer cells, and cancer-unrelated cells are much less so with regard to cancer therapeutics (i.e., anti-tumor or not) in disparate clinical settings.

Remember, CSC may differentiate into non-CSC. Thus, there is a continuum of PD-L1 expression in a totem pole of progenitor stem-like cells and progeny differentiated cells. Importantly, this difference in immune checkpoints supports if not proves a stem cell origin of cancer (e.g., hierarchy and heterogeneity) and its implications on immunotherapy in cancer care.

### 4.4. Benefits vs. Risks (What?)

A mandate for doctors in patient care is to cause no harm. A caveat to maximize benefit and minimize toxicity in cancer care is to target cancer cells but spare normal cells. However, if cancer cells share certain properties with normal cells, then the task becomes harder and the risk higher.

Unfortunately, if cancer has a stem cell origin, we may not be able to completely separate cancer stem cells from normal stem cells. The therapeutic ratio tends to be smaller and the therapeutic window narrower, especially when we become overconfident with the power and zeal in the promises of immunotherapy and start to treat less-threatening tumors in those patients who are more likely to be harmed by the treatments than from their cancers.

Therefore, what we treat can make all the difference in terms of efficacy and toxicity with immunotherapy—whether we treat different tumor subtypes or phenotypes; distinct progenitor or progeny cancer cells; disparate tumor compartment, components; or the microenvironment, according to the stem cell theory of cancer [5,6].

Otherwise, we may cause more harm than good. Unfortunately, when we misconceive and misunderstand cancer immunity, when we misplace and misuse immunotherapy, we may instigate hyper-progression of cancer and exacerbate autoimmune complications from immunotherapy.

For example, Kato et al. [38] revealed that 8 (5%) out of 155 patients who had been relatively stable before anti-PD-L1 immunotherapy declined rapidly within 2 months of treatment. Six had their tumors enter a hyperactive state, in which the tumors grew between 53 and 258%.

Champiat et al. [39] showed that 12 (9%) out of 131 patients experienced hyper-progressive disease after anti-PD1/PD-L1 immunotherapies.

Similarly, Ratner et al. [40] reported hyper-progression in several patients with adult T-cell leukemia-lymphoma after immunotherapy. One patient had survived more than 20 years with an indolent form of this cancer with various treatments. Less than one week after one infusion of anti-PD-L1 immunotherapy, her skin lesions turned swollen and warm. Her spleen became massively enlarged and painful. There was a 63-fold increase in her levels of DNA from the cancer-causing virus. She received radiation therapy to shrink her spleen and skin lesions. Her condition improved, but her disease worsened. She died a few months later.

Understandably, there is a sense of unease and urgency (as well as of hubris and denial) when we cannot distinguish hyper-progression from pseudo-progression, in which the scans suggest apparent tumor growth but the tumor is in fact being infiltrated and becoming engorged by armies of immune and inflammatory cells, which occurs in about 10% of patients with melanoma on immunotherapy.

It would be tragic to assume pseudo-progression for hyper-progression and continue a highly deleterious rather than a presumed salacious treatment for such patients.

Perhaps it is not mere coincidence that we observe numerous autoimmune phenomena in patients with malignancies and we diagnose malignant tumors with increasing frequency in autoimmune conditions [41,42].

Assuming that a malignant cell is derived from a stem cell, we postulate that effective immunotherapy against cancer cells may also cause irreparable damage to the host’s stem cells. Presumably, injury to stem cells would cause lasting or permanent sequelae, whereas injury to differentiated cells is repairable or reversible. Indeed, autoimmunity often accompanies successful immunotherapy of some cancers [43,44].

For example, Franzke et al. [43] reported that a positive thyroid autoantibody titer was highly correlated with increased survival in patients with renal cell carcinoma who received systemic IL-2 and IFN-α2 therapy. Similarly, five of six patients who experienced substantial regression of their metastatic melanoma after tumor-infiltrated lymphocyte and IL-2 treatment also developed anti-melanocyte autoimmunity (e.g., vitiligo) [44].

Therefore, effective immunotherapy may elicit an immune response to tumor antigens and to related stem cell antigens. The hypothesis of a stem cell origin of malignancy accounts for the shared antigens and potential cross-reactions between a malignant cell and its stem cell of origin.

### 4.5. Timing and Time (When?)

A stem cell origin of cancer implicates that timing is quintessential. It predicates that time and function are intricately interconnected. A progenitor cancer stem cell follows a different timetable than a progeny differentiated cancer cell does. They have their own unique timelines and deadlines. They act and react differently to the same stimulus and cues at different time points. Their scheduled lineage commitments are distinct. Their circadian metabolic requirements are different.

Consequently, there is a time when therapeutic intervention targeting a specific immunological factor in a certain cell type is appropriate but in another cell type improper. For instance, adult murine neural stem and progenitor cells (NSPC) display increased neuronal differentiation and MHC expression in a STAT-1-dependent manner when exposed to gamma IFN. In contrast, embryonic NSPC exhibit decreased neuronal differentiation and less MHC expression in response to gamma IFN [78,79].

Therefore, phenotypic response to gamma IFN varies depending upon the cell types, i.e., whether it involves progenitor stem cells vs. progeny differentiated cells and cancer stem cells vs. differentiated cancer cells, and whether that response will lead to the recognition of normal cells vs. tumor cells (both of which are supposedly “self” rather than “non-self”) by an intact, competent immune system.

In many respects, integrating treatments that target different cell types and considering different time points is the essence of a stem cell theory and a rationale for multimodal therapy of cancers. Hence, there is a time for preventive measures, when it may be feasible to prevent or delay cancer initiating cells from germinating and sprouting. There is another time for inductive regimens, when it is necessary to palliate debilitating symptoms and eliminate differentiated cancer cells that are wildly proliferating and blossoming. There is also time for consolidative therapies to manage predominant persistent/resistant cancer cells and maintenance programs to suppress minimal residual cancer stem cells from regenerating and disseminating.

In other words, we do not expect that there is a remedy for all cancers all the time. There is a time when treating a primary tumor is obligatory and another time when treating metastatic lesions is preferable. A regimen that controls indolent progenitor cancer stem cells for the purpose of preventing cancer or maintaining a remission is necessarily distinct and different from one that counters fulminant progeny differentiated cancer cells.

Therefore, time is an indelible factor in the evolution of cancer, according to a stem cell theory of cancer. It is integral to the design of neoadjuvant therapy, adjuvant therapy, and palliative therapy in a multimodal approach for the management of complex, mixed tumors. There is an optimal time to control systemic disease and another time to eradicate localized disease which may be resistant to the same treatment due to its innate heterogeneity that defies time and denies immunotherapy.

### 4.6. Microenvironment (Where?)

A stem cell origin of cancer implicates that location is paramount. It predicates that where a cell resides defines its destiny and dictates its deportment. A progenitor cancer stem cell occupies a separate space than a progeny-differentiated cancer cell does. However, not only do they intimately interlink but they also closely interact within and without the confines of their respective microenvironment. In many respects, their very identity and activity depend on their unique microenvironment, and vice versa.

A prevalent microenvironment pertinent to the immune system and immunotherapy that has garnered increased scientific attention pertains to the microbiome. How immunotherapy acts on the microbiome and how the microbiome reacts to immunotherapy involving a multitude of host and neighboring cells, cancer and non-cancer cells, progenitor and progeny cells, and how this interrelationship determines efficacy and safety as well as dictates overall patient outcome and well-being is of interest.

When we consider how a healthy gut relates to a healthy body, the connection between food and medicine becomes inevitable. The gut is home to trillions of microorganisms, including bacteria. The food we eat feeds both our body and our bacteria. When we observe that our microbiome affects our gastrointestinal system and immune system, our body habits and mental status, it is hard not to grasp that food affects our health. Like medicine, food can make us well or sick.

When it concerns cancer, certain bacteria may be pro-tumorigenic and others anti-tumorigenic. Bullman et al. [45] found that the persistence of nearly identical Fusobacterial strains in colorectal cancer tissues suggests that bacteria may migrate with colorectal cancer cells to the metastatic site. Similarly, Dejea et al. [46] showed that the colonic mucosa of FAP patients were highly enriched with patchy bacterial biofilms composed predominantly of *Escherichia coli* and *Bacteroides fragilis* that secrete oncotoxins, colibactin and fragilis toxin, respectively.

In contrast, *Akkermansia muciniphila* is a microbe in our gut that is good for our health. It protects us from malignancy and autoimmunity. *A. muciniphila*, as its namesake implies, produces mucin that lines our gut. This mucus layer prevents leaking of undigested food particles and bacteria into the bloodstream that initiates an inflammatory and immune response and instigates malignancy and autoimmunity, respectively. Interestingly, *A. muciniphila* is associated with improved clinical outcome and enhanced anticancer effects of immunotherapy in patients with lung, melanoma, and kidney cancers [47,48,49,50].

Similarly, *Bifidobacterium adolescentis*, *Barnesiella intestinihominis*, and *Clostridium butyricum* have been found to correlate with improved clinical outcome from CPI [51,52,53]. Interestingly, transplantation of fecal material enriched with *Bifidobacterium* spp. alone (even without CPI) was sufficient to delay tumor growth in preclinical models [48,54]. Intriguingly, the benefit of *Clostridium butyricum* was observed to be more pronounced in patients who had received antibiotic therapy, given that antibiotics have consistently been found to diminish the impact of CPI [55].

Importantly, Spencer et al. [80] reported that advanced melanoma patients who consumed at least 20 g of dietary fiber per day lived longer with anti-PD1 therapy. They found that dietary fiber increased the presence of a family of bacteria, *Ruminococcaceae*, in the gut flora and the production of certain short-chain fatty acids, such as propionate, which contributed to positive antitumor effects. Surprisingly, generic dietary fiber rather than specific microbes (in a probiotic) induced a favored immune response and improved clinical outcomes.

Cheng et al. [81] demonstrated a connection between diet which affects the microbiome and intestinal stem cells. The implication of immunotherapy targeting PDL1-bearing cancer stem cells elicits anti-cancer effects but also normal stem cells which results in immune-mediated colitis suggests that modulating the microbiome may enhance therapeutic efficacy and attenuate potential serious and severe toxic effects. It also suggests that what may be beneficial in a preventive setting (protecting the normal stem cells with a “healthy” microbiome) may not be so in a palliative setting (enabling cancer stem cells with an unhealthful microbiome).

Therefore, the idea of the microbiome affecting the behavior of cancer cells and efficacy of cancer therapy emphasizes the effects of the cellular microenvironment and of cell–cell interactions on our health and in cancer care. It espouses the principle of multicellularity and embraces the theory of a stem cell origin of cancer, in which there is unity if not union of a multitude of cells, including cancer, immune, and bacterial cells.

### 4.7. Cancer Theory (How?)

A stem cell theory of cancer has therapeutic implications. How we harness its potential and fulfill its promise in scientific research and clinical practice is the ultimate question. And the elusive answer may be hidden in some elemental questions: How do we assure that our scientific research conforms to the scientific method? How do we ensure that the goals of drug development serve rather than usurp the purposes of therapy development?

When cancer has a stem cell origin and is a stem cell disease, immunotherapy may be paradoxically beneficial for a reason but ineffectual for the same reason. When cancer is a mixed tumor, immunotherapy needs to confront heterogeneity. When cancer is dynamic rather than static and interactive rather than isolated, immunotherapy needs to treat both the whole tumor and its parts, and tame not only the cancer itself but also its niche.

Germ cell tumor of the testis (TGCT) is an ideal tumor model to investigate a stem cell origin of cancer, because of its overt stem-ness and blatant heterogeneity [82,83]. After all, a germ cell is a prototype stem cell. There is an explanation for its exemplary curability, and an excuse for its exceptional immunity.

About half of TGCT is seminoma; the other half is nonseminoma. About 80% of nonseminoma comprises various combinations and permutations of embryonal carcinoma, choriocarcinoma, yolk sac tumor, teratoma, and/or seminoma. In a mixed TGCT, the genetic makeup is similar if not identical among its various components because of a common clonal origin. However, the clinical course of each subtype within the whole tumor cannot be more dissimilar, if not diametrically opposite. Hence, embryonal carcinoma is fulminant and may be exquisitely chemosensitive. In contrast, teratoma is indolent and completely chemoresistant. Therefore, to cure a mixed nonseminoma, we give chemotherapy to eliminate the embryonal carcinoma and we perform surgery after chemotherapy to remove any residual teratoma.

Of interest, 73% of seminoma and 64% of nonseminoma express PD-L1 [84,85]. Perhaps it is understandable that there is a limited role for anti-PD1/L1 therapy given that >90% of TGCT is already curable by surgery, radiation therapy, and/or chemotherapy. However, there is a still glaring gap in our current knowledge about immunotherapy when most patients with refractory, PD-L1+ TGCT do not respond to and may not benefit from anti-PD1/L1 therapy [86].

A gap, perhaps even a chiasm, in our current understanding about cancer immunity pertains to the pertinence of immune infiltrates in the prognostication of TGCT subtypes. Perhaps it will take a unified theory of cancer to close this gap and bridge this chiasm of cancer ignorance.

Hvarness T. et al. [87] performed phenotypic characterization of immune cell infiltrates in testicular germ cell neoplasia. They found that the phenotype of the infiltrating inflammatory cell composition was comparable in testes from infertile men without neoplasia, in testes with a premalignant lesion (i.e., carcinoma in situ), and/or in overt TGCT.

Therefore, immune infiltrate has vastly different meanings depending on the questions we pose and the theory we posit. All seminomas have immune infiltrates, which is a criterion for the diagnosis of seminoma. One may wonder whether the presence of immune infiltrates accounts for the improved prognosis of seminomas. Alternatively, one may ponder whether it is the nature of a distinct tumor subtype (i.e., seminoma), which attracts immune infiltrates that dictates the prognosis. In other words, immune infiltrate could be an effect rather than the cause of malignancy depending on the perspective we espouse and the narrative we expound.

We forewarn that detection of immune infiltrates may not necessarily improve our current prognostic or predictive capabilities in cancer care, unless we adopt the correct scientific theory to elucidate it and adhere to the proper scientific method to investigate it.

And how we postulate the proper scientific theory and formulate the right hypothesis depends on whether they are derived from pertinent observations in the clinics and in nature, or from fantastic discoveries made in the laboratory and mentioned in the textbooks. In other words, we need to make sure that we design experiments to test and not to generate hypotheses, according to the scientific method. Otherwise, we become entrapped and may be enamored with a hypothesis that is likely to be false, flawed, and fallacious.

Therefore, it is imperative that we adopt the scientific method and adhere to its basic principles. A proper scientific theory and pertinent hypothesis will equip and empower us with the right perspective and narrative regarding cancer immunity and immunotherapy. Theoretically, it will enable us to advance scientific research in the right direction to the right destination. Practically, it will separate drug development from therapy development [53,54] and elevate both endeavors to enhance cancer care.

## 5. Conclusions

A simple way to understand the immune system is to separate self from non-self (Figure 1). If it is self, the immune system tolerates and spares. If it is non-self, the immune system eradicates and strikes.

Consequently, if cancer has a stem cell origin and nature, as manifested by its EMT phenotype and PD-L1 expression, we have a serious problem and a major dilemma with immunotherapy. Because most refractory cancers containing cancer stem cells are likely to be more self than non-self, immunotherapy will be an uphill battle and could be a pyrrhic victory in cancer care.

In the ongoing saga of immunotherapy, we are at a critical juncture. Because of the allure and promises of immunotherapy, we will be treating more patients not immediately threatened by their cancer. They may have more to lose than to gain, if we have a misconception and if we are on a wrong mission with immunotherapy.

A stem cell theory of cancer is supposed to be the unified theory of cancer, because it is comprehensive and universal. It considers all cancer hallmarks, including heterogeneity and immunity. It accounts for both cancer genetics and epigenetics. It encompasses multiple cancer compartments, myriad cancer components, and the ubiquitous cancer microenvironment. Knowing the why, whom, which, what, when, where, and how of immunotherapy will strengthen its role, value, and utility in the big scheme of cancer care: as a prominent part of multimodal therapy (versus targeted therapy) to enhance cancer care and as an integral part of integrated medicine (versus precision medicine) to attain a cancer cure.

According to the stem cell theory of cancer, we should be careful with immunotherapy. We do not want to mess with an intricate immune system and the indomitable cancer stem cells. When we do not recognize or realize that cancer originates from a stem cell and has innate stem-ness capabilities, we may cause more harm than good in many patients and fail to separate the truth from the myth about immunotherapy.

## Figures and Tables

**Figure 1 cancers-15-05385-f001:**
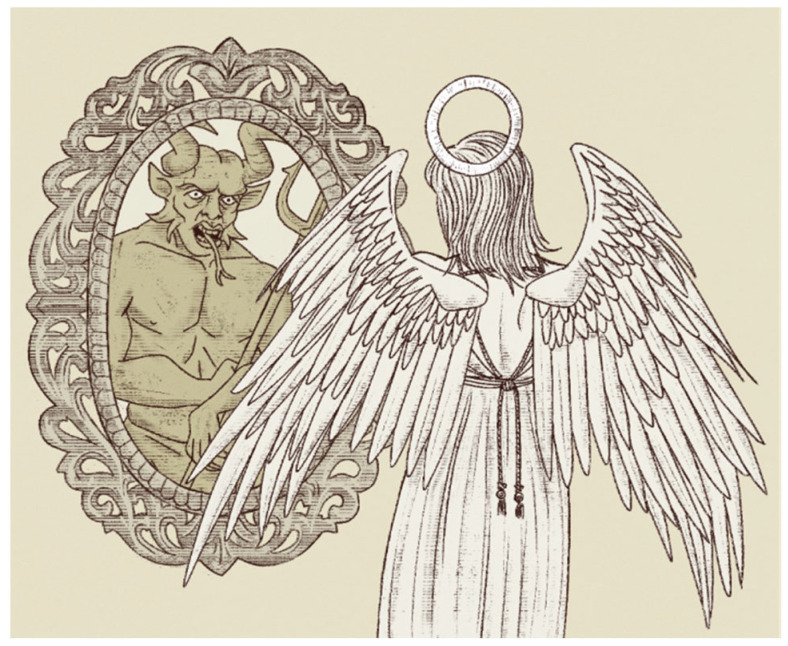
Myth of Lucifer, the fallen angel. Cancer stem cells mirror, if not mimic, normal stem cells. Reproduced with permission from Benjamin Tu (www.bentubox.com, accessed on 10 September 2023).

**Table 1 cancers-15-05385-t001:** Questions and challenges of immunotherapy in cancer care.

Questions and Challenges	Examples (Year, First Published or Approved)	References
WHY? Cancer vaccines	Prostate cancer: Sipuleucel-T (2010); GVAX (2009); PROSTVAC (2019) Melanoma: Peptide (2015); Vaccinia viral lysate (2002); allogeneic melanoma lysate (2007); GM2-KLH/QS-21 (2001); MAGE-A3 (2001) Kidney cancer: MVA-5T4 (2010); IMA901 (2016); autologous tumor cells + BCG (1996); VITESPEN (2009)	[7,8,9,10,11,12,13,14,15,16,17,18,19,20,21,22,23]
WHOM? Patient selection	MSI (2020); TMB (2020)	[24,25]
WHICH? CPIs are not equal CSC vs. non-CSC	Anti-cancer: anti-PD1/L1 (2015) Immune activation: anti-CTLA4 (2013), anti-IDO1 (2019), anti-TIGIT (2022), anti-CD47 (2023), NKTR-214 (2022)	[26,27,28,29,30,31,32,33,34,35,36,37]
WHAT? Benefits vs. risks Tumor subtypes	Hyper-progression (2016) Autoimmunity (1999)	[38,39,40,41,42,43,44]
WHEN? Timing and time	Neoadjuvant (2020) Adjuvant (2020) Inductive/Consolidative (2018) Maintenance (2018)	[5,6]
WHERE? Microenvironment	Microbiome (2015)	[45,46,47,48,49,50,51,52,53,54,55]
HOW? Cancer theory Scientific method	Genetic (1951) vs. stem cell (1863) disease	[5,6]

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
