# Peer review of "Stem Cell Origin of Cancer: Clinical Implications for Cancer Immunity and Immunotherapy"

_cancers, 2023, doi:10.3390/cancers15225385_

Round 1
Reviewer 1 Report (Previous Reviewer 1)
Comments and Suggestions for Authors
the authors addressed my previous comments and suggestions
Author Response
We thank the reviewer for your insightful comments, which have enabled us to enhance the quality of this manuscript.
Reviewer 2 Report (New Reviewer)
Comments and Suggestions for Authors 1. I still think using Figure 1 is not suitable. 2.And the subtiltles such as why, whom,what, where etc. are too short to describe the concept in each section. 3.They should refer to more articles on the relationship between stem cells and cancer stem cells, and try to quote the definitions instead of guessing.Author Response
- I still think using Figure 1 is not suitable.
We believe that some reviewers and many readers may like a subliminal message and agree with the power of myth in Figure 1.
Originally, we considered the myth of Narcissus for “mirror images”. But then we thought that a cancer stem cell is more like Lucifer than Narcissus and commissioned this “original” figure to show Lucifer (the fallen angel) symbolizing cancer stem cell (“malignant devil”) and reflecting normal stem cell (“good angel”) for this Perspective article.
2.And the subtiltles such as why, whom,what, where etc. are too short to describe the concept in each section.
We have added some pertinent details to the subtitles to depict the main concepts for the questions posed in each section, as recommended.
3.They should refer to more articles on the relationship between stem cells and cancer stem cells, and try to quote the definitions instead of guessing.
We defined cancer stem cells in an appendix on page 4, paragraph 3: a small subpopulation of cells within tumors capable of self-renewal, differentiation, and tumorigenicity when transplanted into an animal host.
References 5 and 6 in the introduction of this article comprise two books with a total of about 856 references that address the relationship between stem cells and cancer stem cells, according to the stem cell theory and a unified theory of cancer’s origins and nature.
Round 2
Reviewer 2 Report (New Reviewer)
Comments and Suggestions for Authors
After modification, it has improved a lot.
This manuscript is a resubmission of an earlier submission. The following is a list of the peer review reports and author responses from that submission.
Round 1
Reviewer 1 Report
Comments and Suggestions for Authors
Please find below my comments.
In their narrative the authors seem to be biased exclusively towards their point of view based on classical cancer stem cells hypothesis (“cancer is a stem cell disease”), whereas unbiased approach should provide balanced and pluralistic view on any subject, providing both facts supporting and challenging their hypothesis.
The classical cancer stem cells (CSCs) hypothesis of cancer is not novel (indeed, it's been around for almost a couple of decades), it also has many limitations and can be questioned (https://www.ncbi.nlm.nih.gov/pmc/articles/PMC2593759/; https://www.ncbi.nlm.nih.gov/pmc/articles/PMC6910685/; https://www.ncbi.nlm.nih.gov/pmc/articles/PMC8316871/).
Other models of the cancer origin do exist, for example aforementioned non-hierarchical Stemness Phenotype Model (https://www.ncbi.nlm.nih.gov/pmc/articles/PMC8316871/). The role of the so-called tumour-propagating cells, rather than CSCs, should also be considered when talking about cancer initiation and progression (https://www.nature.com/articles/s41388-019-1128-4).
Many adult stem cells in the human body are considered immunoprivileged.
In Line 95 the authors state that “Understandably, normal stem cells are protected from the rigors of a normal immune system by virtue of its abundant immune checkpoints, its lack of MHC class I molecules, its alliance with HLA-G antigen and regulatory T cells and truce with natural killer cells
and cytotoxic T cells. Unsurprisingly, there will be a predominance of inhibitory immune
cells and a preponderance of inhibitory immune factors in a normal stem-ness microenvi
ronment”. Please support these statements by references to corresponding works.
Line 124. The authors state “However, vaccines have been generally ineffective in cancer therapy” there is a resurgence of vaccines for cancer therapy, several cancer vaccines have been approved by FDA (https://jhoonline.biomedcentral.com/articles/10.1186/s13045-022-01247-x), and the authors even discuss some of them later in the text. However, in my opinion they tend to give a bed rap to the cancer vaccines, focusing on their flaws rather than successes and promises. Once again, I suggest more neutral tone of the narrative and more balanced analysis.
Line 142. “There is debate whether the results of sipuleucel-T represent a statistical sleight
of hand, or a clinical fluke of nature [7]. Perhaps we will never know”. Apart from having rather pessimistic view on the subject that are the arguments supporting the statement that “perhaps we will never know”? There are ongoing clincial tirals and studies on Sipuleucel-T (for example https://classic.clinicaltrials.gov/ct2/show/NCT05751941).
Line 216. The authors state -
“For example, we already know that normal human stem cells possess highly efficient
DNA repair mechanisms that become less efficient upon differentiation”. - please support this statement with references.
Casorelli et al demonstrated higher expression of mismatch repair (MSH2, MSH6,
MLH1, PMS2), base excision repair (AAG, APEX), and O(6)-methylguanine DNA methyl-
transferase in CD34+ stem cells compared with the terminally differentiated CD34- cells
[27].” - high level of expression does not mean higher functional activity and efficiency
Line 225. “Furthermore, stem cells utilize anaerobic glycolytic metabolism (rather than mito-
chondrial or oxidative metabolism), which reduces oxidative stress and DNA damage as
well as cellular injury in general” - the reference supporting this statement is missing.
Stem cells have another mechanism to minimize the chance of replication error: being
quiescent, remaining at the G0 phase of the cell cycle, and having a short cell cycle (G1, S,
G2, and M) – the reference is missing. Not all stem cells are quiescent
Line 235. “Importantly, when DNA damage repair is impaired, stem cells undergo senescence,
cell death, or differentiation, in order to avoid propagation of potentially harmful genetic
mutations and genomic alterations to their offspring cells.
Consequently, in cases of severe or excessive DNA damage, p53 induces apoptosis
or senescence. Furthermore, activation of p53 suppresses pluripotency genes, such as
Nanog, allowing differentiation to proceed”. - references are missing.
Line 355. “Assuming that a malignant cell is derived from a stem cell, we postulate that effective
immunotherapy against cancer cells may also cause irreparable damage to the host’s stem
cells” - any clinical or pre-clinical data to support this statement?
Line 381. “In contrast, embryonic NSPC exhibit decreased neuronal differentiation and less MHC expression in response to gamma IFN [58]”. Reference 58 is a review, not an original work.
Line 410. The authors spend a great deal of time about the impact of microbiome on response to immunotherapy. In my opinion it has nothing to do with stem cells and is completely out of context.
There are multiple self-quotations throughout the text (Tu SM. et al). Such self-quotation can not be justified by the uniqueness of the quoted works, as they are review articles similar to many other reviews reflecting on a similar subject.
From my point of view the only take home messages of this work are: there is a stem cell hypothesis of cancer, different types of tumors may respond to immunotherapy in different ways, perhaps cancer stem cells may contribute to such response and thus need to be studied in such context. Thus, I see no direct and immediate clinical value of this text neither the direct practical guidance for translational researchers. However, the work summarizes nicely and concisely some recent info on cancer vaccines and cancer microbiome.
Author Response
We thank the reviewers for your insightful comments and suggestions. We have provided our responses below in bold with changes highlighted in red underneath the reviewer comments. We have incorporated all changes recommended by the reviewers with the changes highlighted in red in the manuscript.
Please find below my comments.
In their narrative the authors seem to be biased exclusively towards their point of view based on classical cancer stem cells hypothesis (“cancer is a stem cell disease”), whereas unbiased approach should provide balanced and pluralistic view on any subject, providing both facts supporting and challenging their hypothesis.
The classical cancer stem cells (CSCs) hypothesis of cancer is not novel (indeed, it's been around for almost a couple of decades), it also has many limitations and can be questioned (https://www.ncbi.nlm.nih.gov/pmc/articles/PMC2593759/; https://www.ncbi.nlm.nih.gov/pmc/articles/PMC6910685/; https://www.ncbi.nlm.nih.gov/pmc/articles/PMC8316871/).
Other models of the cancer origin do exist, for example aforementioned non-hierarchical Stemness Phenotype Model (https://www.ncbi.nlm.nih.gov/pmc/articles/PMC8316871/). The role of the so-called tumour-propagating cells, rather than CSCs, should also be considered when talking about cancer initiation and progression (https://www.nature.com/articles/s41388-019-1128-4).
We thank the reviewer for reminding us to make sure that we provide a balanced and pluralistic view on any controversial subject, such as about a stem cell origin of cancer. This perspective article is designed to revisit an alternative point of view regarding the origin and nature of cancer in the context of immunotherapy. We have amended the introduction, adopted a more neutral tone, provided a more balanced analysis, and included the appropriate references in our narrative, as recommended.
Page 2, lines 23-39:
Nowadays, there is controversy about a stem cell theory of cancer [1-4]. A stem cell theory may provide us with a comprehensive understanding and knowledge about the origin and nature of cancer [5,6]. It may be the elusive unified theory of cancer that elucidates the origin of all cancer hallmarks, including heterogeneity and immune evasion [5,6]. It embraces the genomics and epigenomics of cancer. It unites various compartments, different components, and the microenvironment of cancer. Importantly, a unified theory of cancer may empower us to treat the whole rather than just a part of cancer. It advocates multimodal therapy over targeted therapy to upgrade cancer care. It advances integrated medicine over precision medicine to fulfill cancer cure.
Currently, immunotherapy is immensely popular and overwhelmingly prevalent in cancer care. Although its scientific merits are undoubted and clinical values unquestioned, there are some inconvenient truths about its promises and limitations that need to be addressed, so that we do not befall to herd mentality or peer pressure in our scientific endeavors and clinical odysseys. We hope that some burning questions about and an alternative view of immunotherapy will invite informed debate and draw objective attention to its putative shortcomings and drawbacks that will further advance and enhance its clinical utility and benefits in cancer care.
Vezzoni L, Parmiani G. Limitations of the cancer stem cell theory. Cytotechnology 2008; 58:3-9.
Bartram I, Jeschke JM. Do cancer stem cells exist? A pilot study combining a systematic review with the hierarchy-of-hypotheses approach. PLoS One 2019; 14:e0225898.
Kaushik V, Kulkarni Y, Felix K, Azad N, Iyer AK V, Yakisich JS. Alternative models of cancer stem cells: The Stemness phenotype model, 10 years later. World J Stem Cells 2021; 13:934-43.
Vessoni AT, Filippi-Chiela EC, Lenz G, Zirnberger Batista LF. Tumor propagating cells: drivers of tumor plasticity, heterogeneity, and recurrence. Oncogene 2020; 39:2055-68.
Many adult stem cells in the human body are considered immunoprivileged.
In Line 95 the authors state that “Understandably, normal stem cells are protected from the rigors of a normal immune system by virtue of its abundant immune checkpoints, its lack of MHC class I molecules, its alliance with HLA-G antigen and regulatory T cells and truce with natural killer cells and cytotoxic T cells. Unsurprisingly, there will be a predominance of inhibitory immune cells and a preponderance of inhibitory immune factors in a normal stem-ness microenvironment”. Please support these statements by references to corresponding works.
Agudo J, Park ES, Rose SA, Alibo E, Sweeney R, Dhainaut M, Kobayashi KS, Sachidanandam R, Baccarini A, Merad M, et al. Quiescent tissue stem cells evade immune surveillance. Immunity 2018; 48:271-85.
Hass R, Otte A. Mesenchymal stem cells as all-round supporters in a normal and neoplastic microenvironment. Cell Commun Signal 2012;10;26
Line 124. The authors state “However, vaccines have been generally ineffective in cancer therapy” there is a resurgence of vaccines for cancer therapy, several cancer vaccines have been approved by FDA (https://jhoonline.biomedcentral.com/articles/10.1186/s13045-022-01247-x), and the authors even discuss some of them later in the text. However, in my opinion they tend to give a bed rap to the cancer vaccines, focusing on their flaws rather than successes and promises. Once again, I suggest more neutral tone of the narrative and more balanced analysis.
We have amended the introduction, adopted a more neutral tone, and provided a more balanced analysis in our narrative, as recommended. Importantly, we hope that objectivity will ensure neutrality and balance in our narrative in case what is considered neutral and balanced may be in the mind of the believer and in the eyes of the beholder. For example,
Page 5, line 137: “Perhaps we tend to forget or ignore a plethora of negative cancer vaccine trials.”
Page 5, lines 138-139: “But negative studies may have important lessons, messages, and warnings for us all, too.”
Page 5, lines 144-148“Consider the case of sipuleucel-T, the first cancer vaccine ever approved by the FDA in 2010 for patients with asymptomatic or minimally symptomatic metastatic castration-resistant prostate cancer (CRPCa). Statistically, patients who received the vaccine experienced a significant overall survival benefit of about 4 months compared with those who did not.”
Page 5, lines 155-157: “For those asymptomatic or minimally symptomatic CRPCa patients with limited bone or soft tissue metastasis who received sipuleucel-T, is the reported overall survival of 25 months better than expected?”
When cancer vaccines and immunotherapy are overwhelmingly popular and prevalent, asking some inconvenient questions (which nobody dares to ask) about its promises and limitations may perhaps balance any potential imbalance, if not bias, for their applications and implementations in cancer care. We hope that an alternative view will invite informed debate and draw objective attention to their putative shortcomings and drawbacks that will further advance and improve their clinical utility and value in cancer care.
Line 142. “There is debate whether the results of sipuleucel-T represent a statistical sleight of hand, or a clinical fluke of nature [7]. Perhaps we will never know”. Apart from having rather pessimistic view on the subject that are the arguments supporting the statement that “perhaps we will never know”? There are ongoing clincial tirals and studies on Sipuleucel-T (for example https://classic.clinicaltrials.gov/ct2/show/NCT05751941).
Currently, most of us in the clinics do not give sipuleucel-T for our patients, because it is difficult to determine benefits (PSA does not decrease, unable to palliate patients when they are asymptomatic) and because there are other treatment options. Surely, drug companies will sponsor more clinical trials to promote and justify the utility of their products. If the results turn out to be negative like those from the GVAX and PROSTVAC trials, then it will be our responsibility to learn why sipuleucel-T succeeds or fails. Unfortunately, when a treatment fails, there will be less incentive (less funding) or motivation (purpose of this perspective to instigate interest) to learn from a negative experience as well as from an unpopular stance.
Line 216. The authors state - “For example, we already know that normal human stem cells possess highly efficient DNA repair mechanisms that become less efficient upon differentiation”. - please support this statement with references.
Casorelli I, Pelosi E, Biffoni M, Cerio AM, Peschle C, Testa U, Bignami M. Methylation damage response in hematpoietic progenitor cells. DNA Repair 2007;6:1170-8.
Bracker TU, Giebel B, Spanholtz J, Sorg UR, Klein-Hitpass L, Moritz T, Thomale J. Stringent regulation of DNA repair during human hematopoietic differentiation: a gene expression and functional analysis. Stem cells 2006;24:722-30.
Casorelli et al demonstrated higher expression of mismatch repair (MSH2, MSH6, MLH1, PMS2), base excision repair (AAG, APEX), and O(6)-methylguanine DNA methyl-transferase in CD34+ stem cells compared with the terminally differentiated CD34- cells [27].” - high level of expression does not mean higher functional activity and efficiency.
You are right! However, higher expression did correlate with higher functional activity and efficiency in this case: “MGMT provides significant protection against MNU toxicity and MGMT and MMR play the expected roles in the MNU sensitivity of these cells.”
Corrected on page 7, line 229: “demonstrated higher expression as well as functional activity and efficiency of…”
Line 225. “Furthermore, stem cells utilize anaerobic glycolytic metabolism (rather than mitochondrial or oxidative metabolism), which reduces oxidative stress and DNA damage as well as cellular injury in general” - the reference supporting this statement is missing.
Warburg O. On the origin of cancer cells. Science 1956;123:309.
Zhou W, Choi M, Margineantu D, Margaretha L, Hesson J, Cavanaugh C, Blau CA, Horwitz MS, Hockenbery D, Ware C, et al. HIF1α induced switch from bivalent to exclusively glycolytic metabolism during ESC-to-EpiSC/hESC transition. EMBO J 2012; 31:2103-16.
Stem cells have another mechanism to minimize the chance of replication error: being quiescent, remaining at the G0 phase of the cell cycle, and having a short cell cycle (G1, S, G2, and M) – the reference is missing. Not all stem cells are quiescent.
Kapinas K, Grandy R, Ghule P, Medina R, Becker K, Pardee A, Zaidi SK, Lian J, Stein J, van Wijnen A, et al. The abbreviated pluripotent cell cycle. J Cell Physiol 2013; 228:9-20.
Line 235. “Importantly, when DNA damage repair is impaired, stem cells undergo senescence, cell death, or differentiation, in order to avoid propagation of potentially harmful genetic mutations and genomic alterations to their offspring cells. Consequently, in cases of severe or excessive DNA damage, p53 induces apoptosis or senescence. Furthermore, activation of p53 suppresses pluripotency genes, such as
Nanog, allowing differentiation to proceed”. - references are missing.
Jaiswal SK, Raj S, DePamphilis ML. Developmental acquisition of p53 functions. Genes 2021; 12:1675.
Lin T, Chao C, Saito S, Mazur SJ, Murphy ME, Appella E, Xu Y. p53 induces differentiation of mouse embryonic stem cells by suppressing Nanog expression. Nat Cell Biol 2005; 7:165-71.
Line 355. “Assuming that a malignant cell is derived from a stem cell, we postulate that effective immunotherapy against cancer cells may also cause irreparable damage to the host’s stem cells” - any clinical or pre-clinical data to support this statement?
Yes! Since both normal MSC and CSC express PDL1, it is expected that anti-PD1/L1 would be efficacious but could also cause long-lasting if not permanent autoimmune complications, which should not occur if the damage is on a differentiated cancer cell with a limited or short life span.
Page 9, line 369-370: “Indeed, autoimmunity often accompanies successful immunotherapy of some cancers” (Franzke, 1999; Dudley, 2002).
Line 381. “In contrast, embryonic NSPC exhibit decreased neuronal differentiation and less MHC expression in response to gamma IFN [58]”. Reference 58 is a review, not an original work.
Included original work: Ahn J, Lee J, Kim S. Interferon-gamma inhibits the neuronal differentiation of neural progenitor cells by inhibiting the expression of neurogenin2 via the JAK/STAT1 pathway. Biochem Biophys Res Commun 2015; 466:52-9.
Retained review for the sake of context and comprehensiveness.
Line 410. The authors spend a great deal of time about the impact of microbiome on response to immunotherapy. In my opinion it has nothing to do with stem cells and is completely out of context.
Cheng et al (2019) demonstrated a connection between diet which affects the microbiome and intestinal stem cells.
"Ketone bodies become highly induced in the intestine during periods of food deprivation and play an important role in the process of preserving and enhancing stem cell activity.
"When food isn't readily available, it might be that the intestine needs to preserve stem cell function so that when nutrients become replete, you have a pool of very active stem cells that can go on to repopulate the cells of the intestine."
Page 11, lines 468-475:
The implication of immunotherapy targeting PDL1-bearing cancer stem cells which elicits anti-cancer effects but also normal stem cells which results in immune-mediated colitis suggests that modulating the microbiome may enhance therapeutic efficacy and attenuate potential serious and severe toxic effects. It also suggests that what may be beneficial in a preventive setting (protecting the normal stem cells with a “healthy” microbiome) may not be so in a palliative setting (enabling cancer stem cells with an unhealthful microbiome).
Cheng CW, Biton M, Haber AL, Gunduz N, Eng G, Gaynor LT, Tripathi S, Calibasi-Kocal G, Rickelt S, Butty VL, et al. Ketone Body Signaling Mediates Intestinal Stem Cell Homeostasis and Adaptation to Diet. Cell 2019; 178:1115-31.
There are multiple self-quotations throughout the text (Tu SM. et al). Such self-quotation cannot be justified by the uniqueness of the quoted works, as they are review articles similar to many other reviews reflecting on a similar subject.
It is difficult to convey the clinical implications of a stem cell theory of cancers “globally” in a single topic of immunity and immunotherapy when it is also implicated in metastasis, heterogeneity, dormancy, drug resistance, metabolism, etc.
We hope that a unified theory of cancer will serve this purpose by uniting all the various cancer hallmarks in the 2 books (refs 5, 6) with abundant references for those readers who may be interested in this specific perspective and narrative.
We have removed our 3 review articles with a similar theme pertaining to this specific subject of a stem cell/unified theory of cancer, as recommended.
From my point of view the only take home messages of this work are: there is a stem cell hypothesis of cancer, different types of tumors may respond to immunotherapy in different ways, perhaps cancer stem cells may contribute to such response and thus need to be studied in such context. Thus, I see no direct and immediate clinical value of this text neither the direct practical guidance for translational researchers. However, the work summarizes nicely and concisely some recent info on cancer vaccines and cancer microbiome.
We appreciate the reviewer’s expert insights and opinions! We understand and respect the merits and values of the prevailing viewpoints. However, some of the critical questions posed in this article need to be answered and the issues raised need to be addressed, precisely because they have direct and immediate clinical relevance.
When we do not formulate a proper or correct hypothesis of cancer’s origins and nature (stem-like or otherwise), its consequences on the efficacy and safety of cancer care are not only direct and immediate, but also urgent and far-reaching. For example:
Page 7, lines 278-280: “if many of the other CPI, such as those targeting IDO1, TIGIT, or CD47, are either minimally effective or not effective at all, then we may be wasting our precious resources, energy, and time.”
Page 8, lines 331-333: “and start to treat less threatening tumors in those patients who are more likely to be harmed by the treatments than from their cancers.”
Unfortunately, we are constantly observing and managing serious and severe life-threatening immune-mediated toxic effects and complications in our clinics and in the wards on an almost daily basis. Hopefully, an alternative perspective on the merits and values of immunotherapy will make the treatment even better and much safer.
We added “Clinical” to “Implications” in the title to emphasize this very important point of “direct and immediate clinical value” in this perspective article.
Reviewer 2 Report
Comments and Suggestions for Authors
This is a perspective paper on the immunotherapy. Although, in the title, they put "Stem Cell Origin of Cancer", the contents are focused on the immuno-therapy and immuno-checkpoints. Since there is no logic mentioned or speculated between cancer stem cells and normal stem cells, they should reconsider the title because the current title is misleading. It appears too descriptive to use the figure 1 because the "self or non-self" is an old subject. And the subtiltles such as why, whom, whhat, etc. are too short, even they are put in Table 1, to describe the concept in each section.
The authors should reconsider the structure and entire structure of the perspective, of which concept should be made clear. The current form of the paper is not acceptable for Cancers.
Author Response
We thank the reviewers for your insightful comments and suggestions. We have provided our responses below in bold with changes highlighted in red underneath the reviewer comments. We have incorporated all changes recommended by the reviewers with the changes highlighted in red in the manuscript.
Reviewer 2
This is a perspective paper on the immunotherapy. Although, in the title, they put "Stem Cell Origin of Cancer", the contents are focused on the immuno-therapy and immuno-checkpoints. Since there is no logic mentioned or speculated between cancer stem cells and normal stem cells, they should reconsider the title because the current title is misleading. It appears too descriptive to use the figure 1 because the "self or non-self" is an old subject. And the subtiltles such as why, whom, whhat, etc. are too short, even they are put in Table 1, to describe the concept in each section.
The authors should reconsider the structure and entire structure of the perspective, of which concept should be made clear. The current form of the paper is not acceptable for Cancers.
Hopefully, the logic between cancer stem cells and normal stem cells will be evident to the reviewer in the following paragraphs:
Page 2-3, lines 65-70: “It is evident that there is a close relationship between oncogene and pro-oncogene and a close resemblance between cancer stem cells and normal stem cells. However, it is not yet apparent if this relationship is pivotal to solving the puzzle of cancer and if a resemblance is the missing key to unlocking the origin of cancer. When a proto-oncogene is already existent, perhaps an oncogene is a reinvention. When a normal stem cell is always present, perhaps a cancer stem cell is a recapitulation.”
Page 4, lines 111-115: “Unfortunately, if cancer has a stem cell origin and is a stem cell disease, then whether cancer stem cell is self-or non-self not only presents us with some perturbing and pressing questions, but also confronts us with some perplexing and profound challenges when it concerns cancer immunity and cancer care. If the immune system spares normal stem cells, then it will also exempt cancer stem cells.”
Perhaps a link between “Stem Cell Origin of Cancer” and “Cancer Immunity and Immunotherapy” that may be lost to the reviewer is in the word “Implications” and may be clarified by the addition of “Clinical” to “Implications”, since this is a perspective from clinicians who treat cancer patients with immunotherapy and who would like to learn about the origin and nature of the disease as well as the merits and values of its treatments, but who also would like to ask some basic questions about the efficacy and safety of such treatments.
Considering the overwhelming attention given to the promises of cancer vaccines and immunotherapy (deservedly so), we believe that one way to improve what we do and what we have today in the field of cancer immunotherapy is to ask some inconvenient questions (which nobody dares to ask) about a putative connection between cancer stem cells and normal stem cells that need to be addressed to further advance cancer research and therapy in the immediate future.
Although the structure is simplified, we believe that it is detailed enough to serve this important purpose of asking some critical questions and facilitating the discussions about a complex subject with brevity and clarity that may appeal to a general readership which includes scientists, clinicians, and layperson.
It is true that “self or non-self” is an old subject. However, whether cancer stem cell is “self or non-self” not only presents us with some perturbing and pressing (new) questions, but also confronts us with some perplexing and profound challenges when it concerns cancer immunity and cancer care (amended on page 4, lines 111-114).
Page 12, lines 553: “Because many refractory cancers containing cancer stem cells are likely to be more self than non-self, immunotherapy will be an uphill battle…”